# A Novel R2R3-MYB Transcription Factor *Sb*MYB12 Positively Regulates Baicalin Biosynthesis in *Scutellaria baicalensis* Georgi

**DOI:** 10.3390/ijms232415452

**Published:** 2022-12-07

**Authors:** Wentao Wang, Suying Hu, Jing Yang, Caijuan Zhang, Tong Zhang, Donghao Wang, Xiaoyan Cao, Zhezhi Wang

**Affiliations:** 1National Engineering Laboratory for Resource Development of Endangered Crude Drugs in Northwest China, Key Laboratory of the Ministry of Education for Medicinal Resources and Natural Pharmaceutical Chemistry, Shaanxi Normal University, Xi’an 710062, China; 2University of Chinese Academy of Sciences, Beijing 100049, China; 3Key Laboratory of Plant Resources Conservation and Sustainable Utilization, South China Botanical Garden, Chinese Academy of Sciences, Guangzhou 510650, China; 4National Maize Improvement Center, College of Agronomy and Biotechnology, China Agricultural University, Beijing 100094, China

**Keywords:** *Scutellaria baicalensis*, R2R3-MYB, SbMYB12, flavonoids biosynthesis, baicalin, positive regulation

## Abstract

*Scutellaria baicalensis* Georgi is an annual herb from the *Scutellaria* genus that has been extensively used as a traditional medicine for over 2000 years in China. Baicalin and other flavonoids have been identified as the principal bioactive ingredients. The biosynthetic pathway of baicalin in *S. baicalensis* has been elucidated; however, the specific functions of R2R3-MYB TF, which regulates baicalin synthesis, has not been well characterized in *S. baicalensis* to date. Here, a S20 R2R3-MYB TF (*Sb*MYB12), which encodes 263 amino acids with a length of 792 bp, was expressed in all tested tissues (mainly in leaves) and responded to exogenous hormone methyl jasmonate (MeJA) treatment. The overexpression of *SbMYB12* significantly promoted the accumulation of flavonoids such as baicalin and wogonoside in *S. baicalensis* hairy roots. Furthermore, biochemical experiments revealed that *Sb*MYB12 is a nuclear-localized transcription activator that binds to the *SbCCL7-4*, *SbCHI-2*, and *SbF6H-1* promoters to activate their expression. These results illustrate that *SbMYB12* positively regulates the generation of baicalin and wogonoside. In summary, this work revealed a novel S20 R2R3-MYB regulator and enhances our understanding of the transcriptional and regulatory mechanisms of baicalin biosynthesis, as well as sheds new light on metabolic engineering in *S. baicalensis*.

## 1. Introduction

In eukaryotes, transcription factors (TFs) play a critical role in plant development, growth, and stress responses through the self-regulation and control of the expression of target genes [1]. A class of transcription factors referred to as MYB contains the MYB domain, which was first identified in maize and is involved in anthocyanin biosynthesis [2]. Myeloblastosis (MYB) proteins can be divided into the R1-, R2R3-, R1R2R3- (3R-), and 4R-MYB proteins [2,3], with the R2R3-MYB family of higher plants being unique and the largest in size. Recently, whole-genome sequencing has enabled the identification of the *R2R3-MYB* gene family in a variety of plants (e.g., *Arabidopsis thaliana* [4], *Populus trichocarpa* [5], *Brassica napu* [6], *Zea mays* [7], *Ananas comosus* [8], *Oryza sativa* [9], *Hypericum perforatum* [10], etc.). These genes influence a variety of processes including the regulation of biotic and abiotic stress responses, secondary metabolites, plant growth and development, as well as cell fate and identity [11,12].

Flavonoids are ubiquitous in nature and extensively distributed in various plant parts. Studies have revealed that flavonoids are highly conserved in nature, with their synthetic pathway being one of the best-known secondary metabolic pathways in plants today [13]. A particular subfamily of *MYB* (*R2R3-MYB*) plays a critical role in the positive regulation of flavonoid synthesis, which for *Arabidopsis* includes S6 (*AtMYB11*/*12*/*111*) and S7 (*AtMYB75*/*90*/*114)* [14,15]. Studies have shown that *AtMYB112*, which is a member of the stress-resistant S20 family, can positively regulate the accumulation of anthocyanin in *Arabidopsis* [16].

*Scutellariae* Radix is a traditional Chinese remedy derived from the dried roots of *S. baicalensis*, which is widely used in China. Modern pharmacological studies have shown that baicalin and other flavonoids are the primary pharmacodynamic constituents of *S. baicalensis*, which have antitumor, antiviral, analgesic, and other beneficial pharmacological effects [17]. Thus, it is referred to as the “antibiotic of traditional Chinese medicine” due to its excellent inhibitory impacts on bacteria, fungi, and viruses. The baicalin content of *S. baicalensis* is typically > 10%, which has garnered the attention of many researchers. Recent studies have revealed that baicalin and baicalein can effectively inhibit the replication of the new coronavirus (SARS-CoV-2) [18,19]. The baicalein content of *S. baicalensis* was observed to increase following water stress, and moderate drought can generally promote the accretion of flavonoids. Suspension cultures of *S. baicalensis* cells were shown to stimulate the accumulation of flavonoids following light and PEG treatments. The exogenous administration of hormones was observed to stimulate the accumulation of baicalin, which is an active constituent of Baicalaria [20]. With the publication of the complete *S. baicalensis* genome, the synthetic pathways of baicalin and other flavonoids were also elucidated. In contrast, very few investigations into the molecular kinetics responsible for the synthesis of flavonoids in *S. baicalensis* have been undertaken.

R2R3-MYB transcription factors such as *At*MYB75, *At*MYB90, *At*MYB113, and *At*MYB114 [14,15]; *Md*MYBA, *Md*MYB1, *Md*MYB10, and *Md*MYB110a [21]; and *Sl*MYB12 [22], *St*AN1 [23,24], and *Sm*MYB98 [25] etc., have been shown to play roles in flavonoid biosynthesis. Qi et al. [26] identified 19 *MYB* genes at the transcriptome level and found that *SbMYB2* and *SbMYB7* negatively affected the phenylpropane content in tobacco through the activation of genes involved in flavonoid biosynthesis. However, to date, no studies have investigated the members of *R2R3-MYB* in *S. baicalensis* that participate in the regulation of specific flavonoids such as baicalin and wogonoside.

In one of our earlier studies based on the whole-genome data of *S. baicalensis*, we identified 95 members of the *R2R3-MYB* family, of which the gene expression levels of *SbMYB12* and *SbMYB74* were significantly increased following ABA and MeJA treatments [27]. *AtMYB112,* which belongs to the S20 family in *A. thaliana*, positively regulated the accumulation of anthocyanins and other flavonoids [16]. Furthermore, another S20 *R2R3-MYB* member (*SmMYB98*) was reported to positively regulate the synthesis and accumulation of salvianolic acid in the transgenic roots of *S. miltiorrhiza*. The sequence structure of *SbMYB12* is akin to that of *AtMYB112* and *SmMYB98*, which suggests that these proteins may perform similar functions in regulating secondary metabolites.

To address this knowledge gap, in this study, a novel member of S20 *R2R3-MYB* (*SbMYB12*) was isolated and characterized from *S. baicalensi*s, which strongly responded to the exogenous MeJA treatment. The regulatory properties of *SbMYB12* in overexpressed *S. baicalensis* hairy root cultures revealed that it enhanced the accumulation of baicalin and wogonoside via the upregulation of a series of key structural genes involved in flavonol biosynthesis. To the best of our knowledge, this is the first report to demonstrate that *R2R3-MYB* from *S. baicalensis* has the capability to enhance the accumulation of baicalin and wogonoside.

## 2. Results

### 2.1. Expression Patterns of SbMYB12 in S. baicalensis 

Based on our previous research [27], *SbMYB12* was classified in subgroup S20 with *AtMYB112*, which is responsible for the accumulation of flavonoids and anthocyanins in *Arabidopsis* [16]. Combined with the RNA-seq results of different tissues and qRT-PCR data under different hormone and abiotic stresses, *SbMYB12* attracted our attention, as it appeared to play an important role in the regulation of flavonoid biosynthesis in *S. baicalensis* [27]. In particular, our earlier study revealed that *SbMYB12* was significantly upregulated in response to MeJA stress, with its expression reaching a peak after 3 h and being upregulated ~10-fold [27]. The induction expression profile suggested that *SbMYB12* was responsive to MeJA.

To evaluate the tissue-specific expression patterns of *SbMYB12*, the roots, root pericytes, root phloem, root xylem, stems, leaves, flowers, and calyx of two-year-old *S. baicalensis* plants were assessed by qRT-PCR. Although the expression of *SbMYB12* was verified for all tested tissues, the highest expression occurred in the leaves (Figure 1).

### 2.2. Extraction and Sequence Analysis of SbMYB12

As shown in Appendix A, the full-length *SbMYB12* gene derived from *S. baicalensis* encodes 263 amino acids with a length of 792 bp. Based on the results of the ExPASy ProtParam tool, the encoded protein has a theoretical molecular weight of 30.249 kDa and a theoretical isoelectric point of 6.19. In the amino acid composition of the *Sb*MYB12 protein, Ser (11.8%), Leu (10.3%), Asn (7.6%), Glu (7.2%), Arg (6.5%), and Asp (6.1%) were prominent. There are 35 negatively charged residues (Asp + Glu) and 32 positively charged residues (Arg + Lys). *Sb*MYB12 exhibited an overall average hydropathicity of −0.786, which implies that the protein is hydrophilic. The protein’s instability index (II) was 62.72, meaning that it was unstable. According to the TMHMM and SignalP web software, *Sb*MYB12 lacked signal peptides and transmembrane sites, which suggests that the protein is a non-secretory and/or non-transmembrane protein (Appendix A).

*Sb*MYB12 had a secondary structure that contained 33.08% α helix, 11.03% extended strand, 6.08% turn, and 49.81% random coils, as shown in Figure 2A. Two conserved SANT domains were found in *Sb*MYB12’s amino acid sequence (Figure 2B), which suggests that it is an R2R3-MYB member. Based on the SWISS-MODEL online analysis tool, the tertiary structure of *Sb*MYB12 was predicted as having an HTH structure, which aligned with its secondary structure prediction (Figure 2C).

The determination of the MYB protein family’s evolution in plants (based on phylogenetic analysis of 18 *Sb*MYB12-related genes from different species) revealed that *Sb*MYB12 has higher homology with *Si*MYB62, *Sm*MYB98, *Ha*MYB62, and *Na*MYB62 (Figure 3A). Furthermore, the alignment of *Sb*MYB12 with other S20 R2R3-MYB subgroup proteins from other species revealed that *Sb*MYB12 shares the conserved R2 domain, R3 domain, and WxPRL motif, which characterizes the S20 R2R3-MYB subgroup protein domain (Figure 3B). According to these results, *Sb*MYB12 possesses a highly conserved domain [25,28,29].

### 2.3. Subcellular Localization and Transactivating Assays of SbMYB12

To further elucidate the classification of *SbMYB12*, we amplified the coding region of *SbMYB12* and fused it to the HBT-GFP-NOS and pGBKT7 vectors, respectively. Subcellular localization analysis indicated that *SbMYB12* was specifically localized within the nucleus (Figure 4A), which was consistent with our prediction [27]. Transactivating assays indicated that *SbMYB12* had strong transcriptional activities in a yeast system (Figure 4B), likely functioning as a TF. These results indicate that *SbMYB12* is a functional nuclear-localized transcription activator.

### 2.4. Transgenic Hairy Roots of S. baicalensis Overexpressing SbMYB12

To validate the functions of *SbMYB12* in flavonoid biosynthesis, we obtained three independent *SbMYB12*-overexpressing (OE) transgenic hairy roots via the *A. rhizogenes* A4-strain-mediated transformation method (Appendix A), which were verified at the genomic DNA level via the fluorescence of the DsRed protein and PCR using the *rolB*, *rolC*, and *SbMYB12* specific primers (Figure 5A,B). The results of qRT-PCR revealed that the transcript level of *SbMYB12* in the three OE lines were significantly higher than that of the control (CK) (Figure 5C). OE-1, OE-2, and OE-3 increased by 23.2, 19.2, and 50.1-fold, respectively, compared with the CK; thus, they were selected for further investigation.

### 2.5. SbMYB12 Positively Regulates the Biosynthesis of Flavonoids in S. baicalensis 

Subsequently, we investigated the phenotype of *SbMYB12*-OE hairy roots and the CK. As shown in Figure 6A, the color of the *SbMYB12*-OE lines was quite different from that of the CK, which may have been caused by different concentrations of secondary metabolites. We quantified the concentrations of five flavonoids (baicalin, wogonoside, baicalein, wogonin, and oroxylin A) in the *SbMYB12*-OE lines and the CK (Figure 6). The contents of baicalin, wogonoside, and total flavonoids (the sum of five flavonoids) in the *SbMYB12*-OE transgenic hairy roots were significantly increased. Compared with the CK, the baicalin content in OE-1, OE-2, and OE-3 increased by 3.48-, 4.44-, and 6.03-times, respectively; the content of wogonoside increased by 1.76-, 1.65-, and 2.20-times, respectively; the content of the total flavonoids increased by 2.08-, 2.55-, and 3.36-times, respectively. Conversely, the content of baicalein in OE-1, OE-2, and OE-3 decreased by 2.99-, 1.69-, and 1.83-times, respectively, and wogonin and oroxylin A did not exhibit significant changes. Overall, these results demonstrate that *SbMYB12* is a positive regulator for flavonoid biosynthesis, especially for baicalin and wogonoside.

### 2.6. SbMYB12 Directly Activated Enzyme Genes of Flavonoid Biosynthesis Pathways 

To reveal the molecular mechanisms behind the *SbMYB12* regulation of flavonoid biosynthesis, we analyzed the transcriptional levels of 24 enzyme genes involved in the flavonoid synthesis pathway via qRT-PCR (Figure 7). Most of these genes, including *SbPAL2*, *SbPAL3*, *SbCCL7-4*, *SbCCL7-5*, *Sb4CL-2*, *Sb4CL-3*, *SbCHI-2*, *SbFNS-2*, *SbF6H-1*, *SbF6H-2*, and *SbUGT*, were significantly upregulated in the *SbMYB12*-OE lines at the *p* < 0.05 level. Of these genes, the expression levels of *SbCCL7-4*, *Sb4CL-3*, *SbCHI-2*, *SbF6H-1*, *SbF6H-2*, and *SbUGT-1* were significantly upregulated at the *p* < 0.01 level. The other enzyme genes had no significant changes in the transgenic lines. These results confirm that *SbMYB12* positively regulates the accumulation of flavonoids such as baicalin and wogonoside by activating its biosynthetic pathway.

MYB proteins typically bind to the cis-elements of enzyme gene promoters to activate or repress their expression to regulate the biosynthesis of secondary metabolites. To further investigate the potential mechanism of how *SbMYB12* regulates flavonoid biosynthesis, we analyzed cis-elements in the promoter regions of enzyme genes in the flavonoid biosynthesis pathway. We found MYB-binding elements (e.g., MRE, MBS1, and MBS3) in the promoter regions of *SbCLL7-4*, *SbCHI-2*, and *SbF6H-1*. Y1H assays were performed, which revealed that *SbMYB12* interacts with the promoter regions of *SbCLL7-4*, *SbCHI-2*, and *SbF6H-1* (Figure 8), indicating that *SbMYB12* binds directly to the promoters of these genes and activates their transcription.

## 3. Discussion

As one of the largest TF families in plants, the MYB family was divided into four subfamilies, namely 1R, R2R3, 3R, and 4R-MYBs [30]. Among them, the R2R3- MYB subfamily consisting of the conserved R2 and R3 MYB domains was the largest MYB subfamily. According to current research, the R2R3-MYBs have been well characterized for their functions in plant developmental processes, stress responses, as well as primary and secondary metabolism [2]. For *S. baicalensis*, 95 R2R3-MYB proteins have been identified and categorized into 34 subgroups [27]. Among them, *SbMYB12* was classified into the S20 subgroups. In Arabidopsis, six S20 members (e.g., *AtMYB2*/*62*/*78*/*108*/*112*/*116*) were reported to be primarily involved in stress responses and the regulation of secondary metabolism [2]. For example, *AtMYB2* was responsible for ABA-induced salt and drought response genes [31]; *AtMYB62* regulated phosphate starvation and gibberellic acid biosynthesis [29]; *AtMYB108* was involved in both biotic and abiotic stress responses [32]. Furthermore, *AtMYB112* (S20) was engaged with regulating the accumulation of anthocyanins and flavonoids in the phenylpropanoid biosynthetic pathway [16]. In *S. miltiorrhiza*, *SmMYB9b*-OE (a S20 R2R3-MYB) efficiently increased the tanshinone concentration in *S. miltiorrhiza* transgenic roots [28]. Subsequently, Hao et al. [25] reported that another S20 R2R3-MYB member (*SmMYB98*) positively regulated the synthesis and accumulation of salvianolic acid and tanshinone by activating the transcription of *PAL1*, *GGPPS1*, and *RAS1* in the transgenic roots of *S. miltiorrhiza* [25]. As a member of the S20 family, it was speculated that *SbMYB12* may have functions akin to the homologous genes *SmMYB98* and *AtMYB112*, which are involved in the regulation of secondary metabolites [16,25]. Moreover, our earlier studies revealed that *SbMYB12* responded to different exogenous hormone and abiotic stress treatments, particularly exogenous MeJA, and may play important roles in the regulation of flavonoid biosynthesis in *S. baicalensis*

Since it was speculated that *SbMYB12* may be a key regulator of flavonoid biosynthesis, we conducted further investigations. As described earlier, *SbMYB12* transcripts were expressed in all tested tissues and preferentially expressed in *S. baicalensis* leaves (Figure 1). The length of the *SbMYB12* nucleotide sequence was found to be 792 bp, encoding 263 amino acids that were predicted to produce a relatively hydrophilic protein (Appendix A). According to an analysis of the conserved domains, the *Sb*MYB12 protein contained two SANT-MYB DNA-binding domains (Figure 2), which implies that the protein is a typical R2R3-MYB member [3]. Multiple sequence alignments between the *Sb*MYB12 protein sequence and other reported S20 R2R3-MYB subgroup proteins from other species revealed that they had a high similarity in the conserved sequence, including the R2 and R3 domains and the WxPRL motif, but were completely different in the non-conserved region (Figure 3). This result was consistent with the transcription factor characteristics [33]. Additionally, phylogenetic analysis revealed that *Sb*MYB12 was phylogenetically closest to *Si*MYB62 and *Sm*MYB98 (Figure 3A). 

The flavonoids (e.g., baicalin, wogonoside, baicalein, wogonin, and oroxylin A) were confirmed to be the main active components of *Scutellariae* Radix [34], which possesses significant antibacterial, anti-inflammatory, anti-COVID-19 virus, and anti-HIV activities [18,19,35]. Baicalin and wogonoside are considered as important evaluation indices for the quality of *Scutellariae* Radix [20]. The functions of three *R2R3-MYB* genes from *S. baicalensis* (*SbMYB2*, *SbMYB7*, and *SbMYB8*) were characterized based on a non-reference transcriptome, and transgenic tobacco plants overexpressing *SbMYB2*, *SbMYB7*, or *SbMYB8* exhibited a higher phenylpropane content by activating the expression of flavonoid-biosynthesis-related genes [26,36]. However, the heterologous expression verification in tobacco did not directly confirm that *SbMYBs* participated in the regulation of *S. baicalensis*-specific flavonoids such as baicalin and wogonoside. Of particular note was that the *SbMYB2*, *SbMYB7*, and *SbMYB8* mentioned above were renamed as *SbMYB45*, *SbMYB95*, and *SbMYB18*, respectively, in the latest study based on their sequence similarity [27]. Thus, based on the above comprehensive analysis, we speculated that *SbMYB12* most likely functions as an activator of some of the *S. baicalensis*-specific flavonoid biosynthesis in *S. baicalensis*. However, additional experimental evidence is required to verify this.

Subcellular location results demonstrated that *SbMYB12* was localized within the nucleus (Figure 5A). Transcription factors were functionally related to their localization [12,25,28]. According to these results, *SbMYB12* may act as a transcription factor during transcriptional regulation. Subsequently, transactivation activity assays demonstrated that *SbMYB12* is a nuclear-localized transcriptional activator, which suggests that it has the capacity to regulate the transcription of target genes in the nucleus on its own, as reported previously [16].

To further uncover the role of *SbMYB12* in *S. baicalensis*, we generated transgenic hairy roots that overexpressed *SbMYB12* and found that the concentrations of baicalin, wogonoside, and total flavonoids were significantly increased in the SbMYB12-OE hairy roots (Figure 6). Furthermore, the expression levels of these key flavonoid biosynthesis enzyme genes were significantly upregulated in the *SbMYB12*-OE lines, particularly *SbCCL7-4*, *SbCHI-2*, *SbF6H-1*, *SbUGT-1*, etc. (Figure 7). As reported, these genes are responsible for the biosynthesis of *S. baicalensis*-specific flavonoids [34]. In addition, the Y1H assay indicated that *SbMYB12* could bind to the promoters of the key enzyme genes (*SbCCL7-4*, *SbCHI-2*, and *SbF6H-1*) and activate their expression (Figure 8). Therefore, we concluded that *SbMYB12* is responsible for the biosynthesis of flavonoids through the direct activation of *SbCCL7-4*, *SbCHI-2*, and *SbF6H-1* transcription in *S. baicalensis*. *SbMYB12* is a potential manipulation target for improving the flavonoid content via genetic engineering.

## 4. Materials and Methods

### 4.1. Plant Materials

Sterile *S. baicalensis* seedlings were obtained by using stem segments as explants. We selected vigorously growing *S. baicalensis* seedlings, removed the leaves, sterilized them, followed by inoculation in an MS solid medium with 0.5mg/L NAA and 1.0mg/L 6-BA, and cultivation in an incubator at 22 ± 2 °C (14 h light/10 h dark cycle, 60 ± 5% humidity). When the lateral buds of *S. baicalensis* germinated and grew to 2 cm, they were removed and transferred to an MS solid medium (Solarbio, Beijing, China) without hormones and cultivated in an incubator to obtain sterile seedlings of *S. baicalensis* for the transformation experiments.

Various tissues including roots, pericytes, phloem, xylem, stems, leaves, flowers, and calyx were harvested from two-year-old *S. baicalensis* plants at the germplasm resource garden of Shaanxi Normal University, respectively, for RNA extraction.

### 4.2. Isolation and Cloning of SbMYB12

The total RNA was extracted from three-month-old *S. baicalensis* plantlets using the TIANGEN RNA Prep Pure Plant kit (Tiangen, Beijing, China). TransScript^®^ First-Strand cDNA Synthesis SuperMix (TransGen Biotech, Beijing, China) was used for cDNA synthesis. *SbMYB12* (evm.model.contig76.33) full-length cDNA was employed as the reference sequence, and the primers (Appendix A) were designed with Primer Premier 5. As the resulting PCR products were linked to a pMD19-T vector (TaKaRa, Dalian, China) and confirmed by sequencing, the results were positively determined.

### 4.3. Bioinformatics Analysis of SbMYB12

The ExPASy-ProtParam tool (https://web.expasy.org/protparam/ (accessed on 20 September 2022)) was employed to predict the primary structure of the *Sb*MYB12 protein, as well as its amino acid composition, molecular weight, theoretical isoelectric point (pI), and stability [37]. ProtScale (https://web.expasy.org/protscale/ (accessed on 20 September 2022)) was used to analyze its hydrophobic properties and charge distributions [38]. For analyzing transmembrane domains, the TMHMM server software (https://services.healthtech.dtu.dk/service.php?TMHMM-2.0 (accessed on 20 September 2022)) was used [39], while the SignalP 5.0 Server (https://services.healthtech.dtu.dk/service.php?SignalP-5.0 (accessed on 20 September 2022)) was employed to estimate signaling peptides [40]. Using the SMART website (http://smart.embl-heidelberg.de (accessed on 20 September 2022)) [41], the protein domains of *Sb*MYB12 were predicted, while its tertiary structure was predicted using SWISS-MODEL (https://swissmodel.expasy.org/ (accessed on 20 September 2022)) [42].

The *Sb*MYB12 and 17 members from the S20 subgroup R2R3-MYB were used to construct a phylogenetic tree with MEGA11 (https://www.megasoftware.net/dload_mac_beta (accessed on 20 September 2022)) using the neighbor-joining method (NJ) [43]. DNAMAN v. 9.0 (https://www.lynnon.com/dnaman.html (accessed on 20 September 2022)) was used to perform multiple sequence alignments between *Sb*MYB12 and other MYB TF species. Appendix A lists the amino acid sequences of *Sb*MYB12 and other MYB proteins.

### 4.4. Expression Analysis Using qRT-PCR

Based on previously described procedures [44], the total RNA was isolated and qRT-PCR analysis was performed, using the *SbACT7* gene as a reference [44]. SYBR green qPCR Mix (Vazyme, Nanjing, China) was employed for qRT-PCR with a real-time fluorescence quantitative PCR detection system (Roche LightCycler® 96 Instrument, Basel, Switzerland). Based on the qRT-PCR data, *SbMYB12* and other corresponding genes were calculated using the 2^−ΔΔCt^ method [27]. All primers used for qRT-PCR and plasmid construction can be found in Appendix A.

### 4.5. Subcellular Location Analyses of SbMYB12

To determine the locations of *Sb*MYB12 proteins in cells, *Sb*MYB12-GFP fusion protein expression vectors were developed by amplifying the *Sb*MYB12 coding sequence and fusing it with HBT-GFP-NOS vectors. *Arabidopsis* leaves were cultured under 12 h light/12 h darkness for four weeks, and mesophyll protoplasts were prepared as previously described [45]. The transformed protoplasts were grown at 21 °C for 12 h and observed using a confocal laser microscope (Leica TCS SP5, Wetzlar, Germany).

### 4.6. Transcriptional Activation Assays of SbMYB12

Using a Gateway recombinatorial cloning system [46], the ORF of *SbMYB12* was cloned and integrated into the pGBKT7 (BD) vector. *Saccharomyces cerevisiae* strain AH109 (Weidi Biotechnology, Shanghai, China) was transformed with recombinant vector BD-*Sb*MYB12 and cultured for 2–3 days at 28 °C on SD/-Trp (Coolaber, Beijing, China), then screened on SD/-Trp/-Ade/-His/X-α-gal (Coolaber, Beijing, China) media to assay the transactivation activity.

### 4.7. Overexpression of SbMYB12 in Hairy Roots of S. baicalensis 

The full-length CDS of *SbMYB12* was cloned into the pK7WG2R (with a dsRed marker gene) to generate the overexpression vector pK7WG2R-*Sb*MYB12 using the Gateway recombinatorial cloning system (Invitrogen, Carlsbad, CA, USA) [46]. Transgenic hairy roots were obtained from the stem segment explants of *S. baicalensis* infected by the *Agrobacterium rhizogenes* A4 strain (Weidi Biotechnology, Shanghai, China) containing the pK7WG2R-*Sb*MYB12 as described previously [47]. In parallel, hairy roots infested with the A4 strain were used as a control. The *SbMYB12*-overexpressed hairy roots were confirmed by DsRed protein fluorescence and the presence of *rolB*, *rolC*, and *SbMYB12* at the genomic DNA level, as described earlier [48]. For liquid culturing, 3–5 cm-long hairy roots were inoculated in a 250 mL conical flask that was filled with 50 mL of a B5 liquid medium (Coolaber, Beijing, China) and propagated in a steady temperature shaking incubator (22 °C at 100 rpm). The hairy roots were gathered after two months and used for qRT-PCR and HPLC analysis.

### 4.8. Measurement of Flavonoid Content in Hairy Roots by HPLC

The flavonoid content was measured using previously described techniques [49]. Briefly, the transgenic hairy roots and control, which were cultured in liquid medium for three months, were harvested and naturally dried at room temperature. The dried hairy root samples were powdered and extracted with 70% ethanol for 1 h under ultrasonication (200 W, 40 kHz). Subsequently, the extracts were filtered through a 0.22 µm microporous membrane for HPLC (Thermo Fisher Scientific UltiMate 3000 HPLC, Waltham, MA, USA).

### 4.9. Yeast One-Hybrid Assay

The complete *SbMYB12* CDS was cloned into the BamH I and Xho I (TaKaRa, Beijing, China) sites of the pGADT7(AD) vector to generate AD-*Sb*MYB12. The promoter fragments of *SbCLL7-4*, *SbF6H-1*, and *SbCHI-2* were inserted into the EcoR I and Sac II (TaKaRa, Beijing, China) sites of the pHIS2 vector to generate pHIS2-p*Sb*CLL7-4, pHIS2-p*Sb*F6H-1, and pHIS2-p*Sb*CHI-2, respectively. The Yeast One-Hybrid (Y1H) assay was performed as described in our previous protocols [50].

### 4.10. Statistical Analysis

The SPSS software (version 26.0) was utilized to analyze the differences via one-way ANOVA (followed by Tukey’s comparisons) and Student’s *t*-tests. Each bar represents the mean and standard deviation (SD) of three replicates. The significance of a statistical difference was defined as * *p* ≤ 0.05 and ** *p* ≤ 0.01, respectively.

## 5. Conclusions

For this study, a novel R2R3-MYB subgroup 20 TF from *S. baicalensis* (*Sb*MYB12) was isolated and functionally characterized. Further, *SbMYB12* was found to be a nuclear-localized transcription activator that positively regulates the accumulation of flavonoids, such as baicalin and wogonoside, by directly binding to the promoter regions of the key enzyme genes *SbCCL7-4*, *SbCHI-2*, and *SbF6H-1*. Together, these results enrich the understanding of the function of the *R2R3-MYB* gene in medicinal plants and illustrate the exploitation of *R2R3-MYB* in flavonoid biosynthesis, which provides a feasible strategy for enhancing the baicalin and wogonoside concentrations via MYB proteins in *S. baicalensis*. In addition, *SbMYB12* was also the first transcription factor cloned and verified gene function in the hairy root transgenic system of *S. baicalensis*, which lays the foundation for further research on regulating the functional genes of *S. baicalensis*.

## Figures and Tables

**Figure 1 ijms-23-15452-f001:**
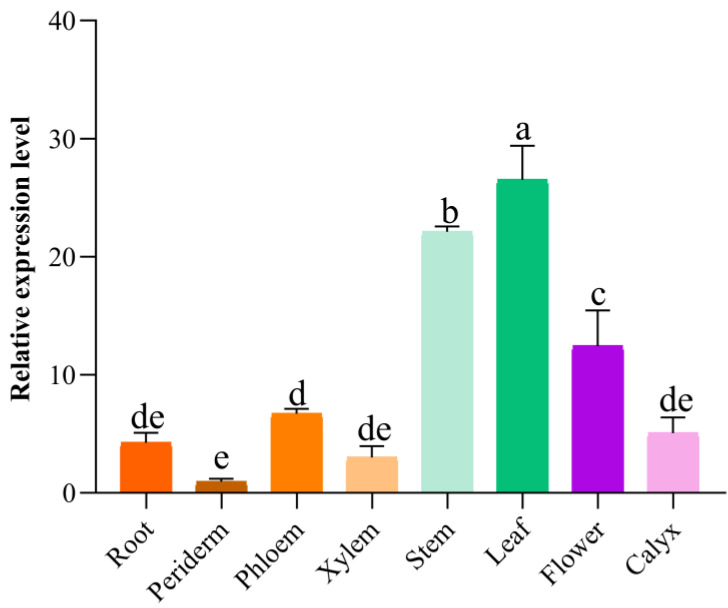
Expression profiles of *SbMYB12* in different tissues of *S. baicalensis*. The data represent the means of three biological replicates, and the error bars indicate the standard deviation (SD). Significant differences between means were identified (depicted by different letters; *p* < 0.05) using one-way ANOVA (followed by Tukey’s comparisons).

**Figure 2 ijms-23-15452-f002:**
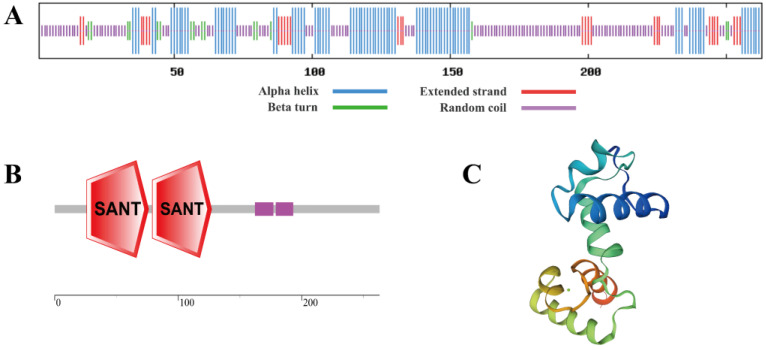
Prediction of *Sb*MYB12 protein structure and domains. (**A**) Predicted protein secondary structure. (**B**) Predicted protein domains. (**C**) Predicted tertiary structure.

**Figure 3 ijms-23-15452-f003:**
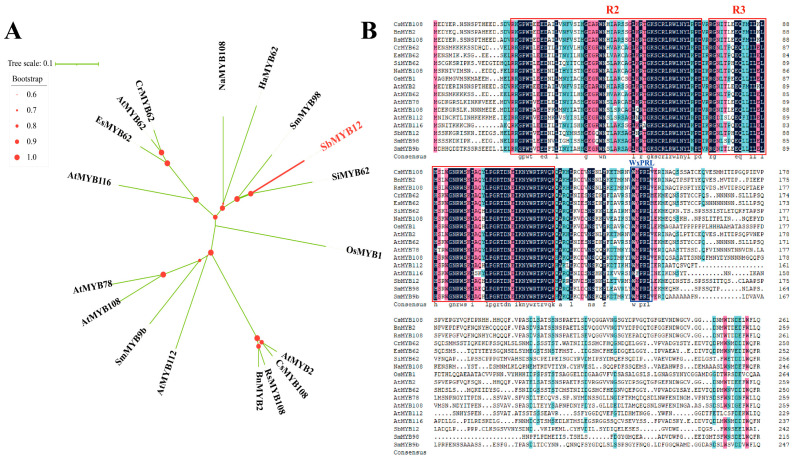
Analysis of *Sb*MYB12 compared with related sequences. (**A**) *Sb*MYB12 gene phylogenetic analysis along with 17 representatives of the R2R3-MYB S20 subgroup. (**B**) Analysis of sequence alignments between *Sb*MYB12 and other R2R3-MYB S20 subgroup proteins from different plants. Red frames indicate conserved R2 and R3 domains, and blue frames indicate the WxPRL core sequence.

**Figure 4 ijms-23-15452-f004:**
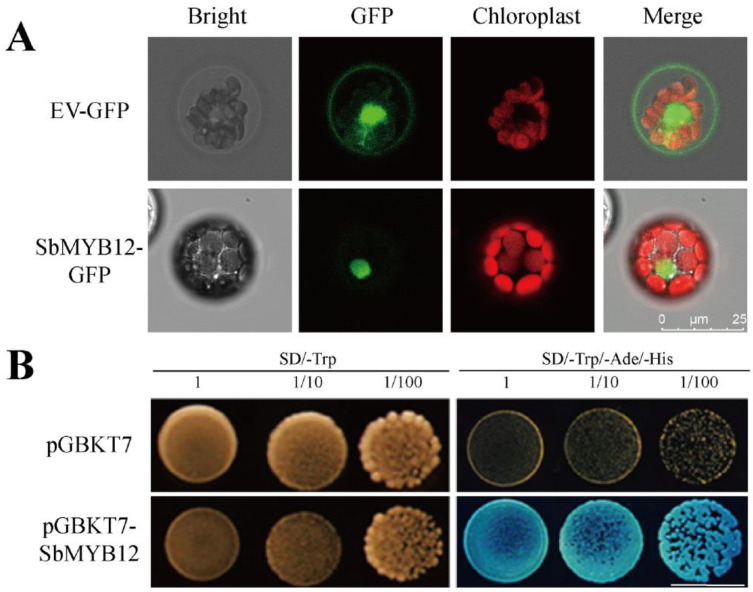
Subcellular localization and transactivation activities of the *Sb*MYB12 protein. (**A**) Subcellular location of *Sb*MYB12 in *Arabidopsis* mesophyll protoplasts. (**B**) Transcriptional activation activity analysis of BD-*Sb*MYB12 in yeast (Scale bar: 8mm).

**Figure 5 ijms-23-15452-f005:**
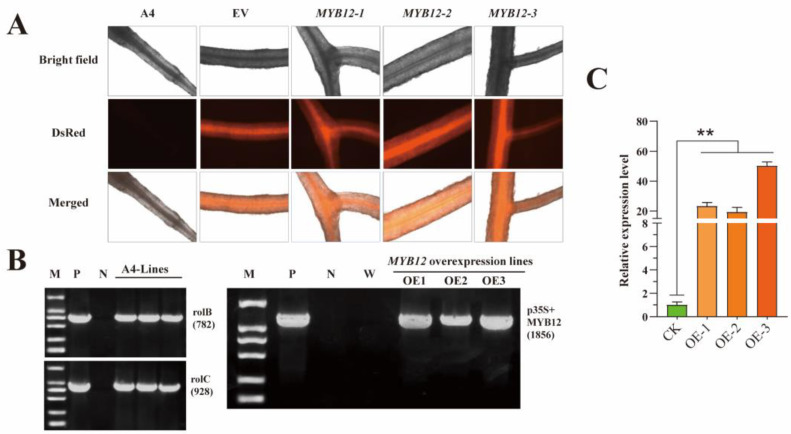
Identification of transgenic hairy roots of *S. baicalensis*. (**A**) Transgenic hairy root lines and observations with red fluorescent protein. A4: ArA4 lines (negative control); EV: ArA4 strain that harbors the pK7WG2R plasmids (positive control); *MYB12*-1~3: *SbMYB12* overexpression hairy root lines. (**B**) PCR identification of transgenic hairy roots of *S. baicalensis*. PCR screening of *SbMYB12* overexpressing lines. M: DNA marker (DL2000); P: ArA4 strain containing recombinant plasmid (positive control); N: *S. baicalensis* sterile plantlet (negative control); W: wild seedling of *S. baicalensis*. (**C**) Real-time quantitative PCR analysis of *SbMYB12* in transgenic lines. CK1-3: ArA4 lines; OE1–3: *SbMYB12* overexpression transgenic line; ** represents significant difference (*p* < 0.01) via Student’s *t*-test.

**Figure 6 ijms-23-15452-f006:**
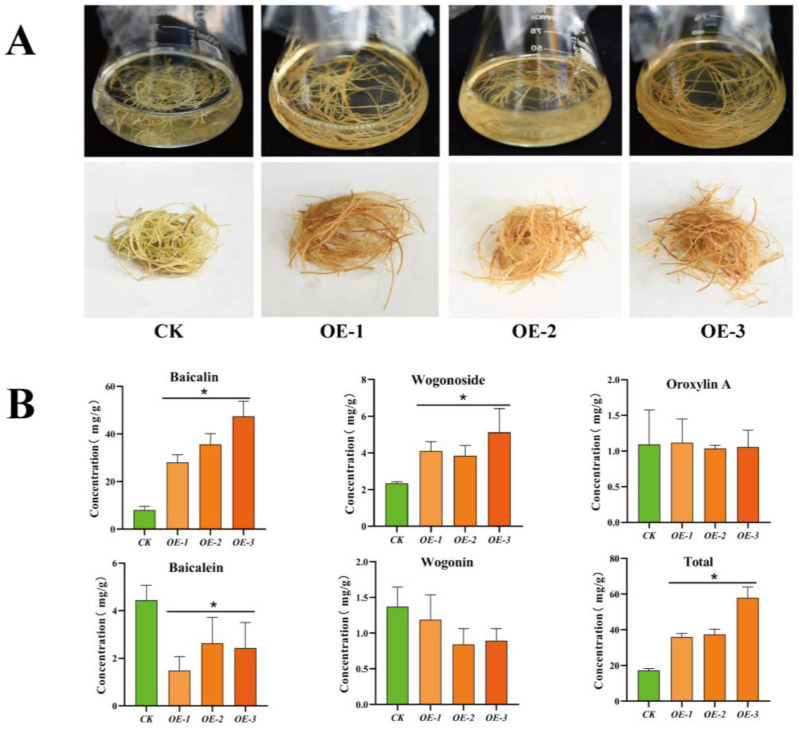
*SbMYB12* promoted the accumulation of flavonoids (baicalin and wogonoside) in *S. baicalensis* transgenic hairy roots. (**A**) Phenotypes of the two-month-old *SbMYB12*-overexpression (OE) lines and the control (CK). (**B**) Concentrations of flavonoids in the transgenic hairy roots and the control (CK). Significant differences between the OE lines and the CK were identified (depicted by * *p* < 0.05) via Student’s *t*-test.

**Figure 7 ijms-23-15452-f007:**
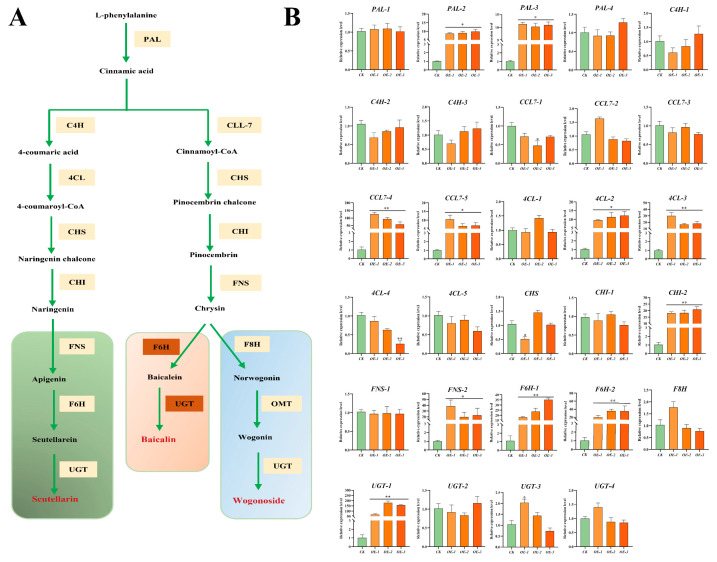
Expression analyses of enzyme genes for the flavonoid biosynthesis pathway in *SbMYB12* overexpressed transgenic hairy roots. (**A**) Proposed biosynthetic pathway for flavonoids in *S. baicalensis*. (**B**) Flavonoid biosynthesis pathway gene expression analysis in S*bMYB12* overexpressed strain. Significant differences between the OE lines and the CK were identified (depicted by * *p* < 0.05, ** *p* < 0.01) via Student’s *t*-test.

**Figure 8 ijms-23-15452-f008:**
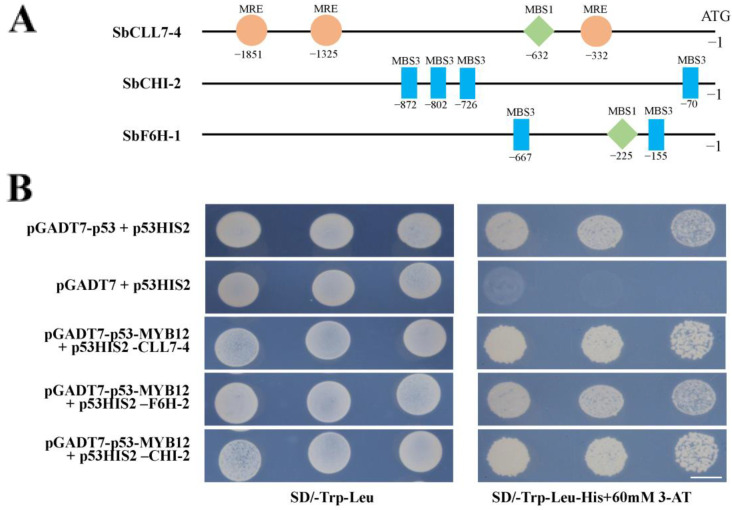
*SbMYB12* binds to the *SbCCL7-4*, *SbCHI-2*, and *SbF6H-1* promoters. (**A**) Distribution of the MYB-binding sites in the *SbCCL7-4*, *SbCHI-2*, and *SbF6H-1* promoters. (**B**) Y1H assays indicated interactions between *SbMYB12* and the promoter regions of *SbCCL7-4*, *SbCHI-2*, and *SbF6H-1*. MRE: AACCTAA; MBS1: CAACTG; MBS3: TAACTG. (Scale bar: 6 mm).

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
