# Peer review of "A Novel R2R3-MYB Transcription Factor SbMYB12 Positively Regulates Baicalin Biosynthesis in Scutellaria baicalensis Georgi"

_ijms, 2022, doi:10.3390/ijms232415452_

Round 1
Reviewer 1 Report
The article is characterized by high innovativeness.
Materials and methodology should be placed before the obtained results.
The authors should describe exactly for what purpose and why they used specific statistical tests. The statistical tests used in the results have not been described in the statistical analysis.
The effect size for the statistical tests used (Cohen's, etc.) should be calculated.
Discussion is not repeating the obtained results - too many repetitions of the obtained results (figures) and too little discussion.
Author Response
Responses to Reviewer 1’s comments
Dear reviewers:
On behalf of all co-authors, I’d like to express our sincere gratitude to you for your time and helpful comments / suggestions on the earlier version of our manuscript (Manuscript ID: ijms-2002576) titled “A Novel R2R3-MYB Transcription Factor SbMYB12 Positively Regulates Baicalin Biosynthesis in Scutellaria baicalensis Georgi”. Your comments are very objective, which is very helpful for the improvement of our manuscript. Along with the revised manuscript with the “Track Changes”, we have included a point-to-point response to your comments. We are also ready to further improve the manuscript if any extra comments are made. Detailed responses to reviewers are given below.
Thanks again for your patient help.
All the best,
Xiaoyan Cao
Reviewer 1: Comments and Suggestions for Authors:
The article is characterized by high innovativeness.
Point 1. Materials and methodology should be placed before the obtained results.
Response 1: Thank you very much for your comment. IJMS encourages authors to use Microsoft Word templates to prepare article submissions. Journal typesetting requires that Materials and Methods of article should be placed behind the Results and Discussions. Therefore, we wrote the article according to the Accepted File Formats.
Point 2. The authors should describe exactly for what purpose and why they used specific statistical tests. The statistical tests used in the results have not been described in the statistical analysis.
Response 2: Thank you for the suggestion. In this article, we have adopted two statistical tests, including Student's t-tests and one way ANOVA (followed by Tukey's comparisons). Different significance analysis methods are applicable to different analysis contents, in which t ‐ tests are applicable to studying the differences between the two groups of samples. Therefore, we choose this analysis method to conduct the differential expression of many genes between SbMYB12 expression (OE) lines and the control (CK). However, one-way ANOVA is often used to analyze the differences between multiple groups of samples. Therefore, we chose one-way ANOVA to analyze the Expression profiles of SbMYB12 in different tissues of S. baicalensis. [Liu, Q., & Wang, L. (2021). t-Test and ANOVA for data with ceiling and/or floor effects. Behavior research methods, 53(1), 264–277.]. In addition, we added the description of the statistical tests in the figure of the results section.
Point 3. The effect size for the statistical tests used (Cohen's, etc.) should be calculated.
Response 3: Thank you very much for your comment. The effect size for the statistical tests used Cohen's d was not considered previously because of the shortcomings of our statistical knowledge. In previous studies, Cohen's d above 0.8 indicates high efficiency, while lower than 0.2 indicates low effect [Bowring, A., et al. (2021). Confidence Sets for Cohen's d effect size images. NeuroImage, 226, 117477.]. According to your suggestion, we have obtained Cohen's d values and effect size of statistical tests for relevant data in this article, and the detailed results are shown in Table 1. It is found that the data supports the P value results in the original manuscript very normatively. Therefore, we continue to maintain the P-value results in the original manuscript in the revision.
Table 1. The effect size for the statistical test
Note: *,* * represent P<0.05 and P<0.01,respectively
Point 4. Discussion is not repeating the obtained results - too many repetitions of the obtained results (figures) and too little discussion.
Response 4: Thank you very much for your comments. According to your suggestion, we have made some changes in the revised manuscript.
Thanks again for your patient help.

Reviewer 2 Report
The manuscript presented describes a novel member of the MYB transcription factors family from S. baicalensis. Authors characterized the gene ant its impact into two main flavonoids of S. baicalensis biosynthesis.
Figure 2 presents levels of expression in different parts of the S. baicalensis plant. From the figure and correspondent description in the text it is not exactly clear xylem, phloem and periderm are the part of what roots or stem?
Also analyzing results presented in figure 2 authors concluded maximal expression levels of SbMYB12 in aboveground part of plant, especially leaves. But usually roots serves as a medicinal raw material for Scutellaria species. And what about concentration of flavonoids, especially baicalin and wogonoside, in native plants? What part of the plant accumulate maximal concentration of flavonoids?
The data on expression profiles of genes supposed to be regulated with SbMYB12 overexpression are obtained on hairy roots model system. So, what is about a sustainability these expression pattern of baicalin synthesis in leaves, steam and roots of intact plant?
A scheme of flavonoid biosynthesis presented in figure 7 is based on authors own of literary data?
Author Response
Responses to Reviewer 2’s comments
Dear reviewers:
On behalf of all co-authors, I’d like to express our sincere gratitude to you for your time and helpful comments / suggestions on the earlier version of our manuscript (Manuscript ID: ijms-2002576) titled “A Novel R2R3-MYB Transcription Factor SbMYB12 Positively Regulates Baicalin Biosynthesis in Scutellaria baicalensis Georgi”. Your comments are very objective, which is very helpful for the improvement of our manuscript. Along with the revised manuscript with the “Track Changes”, we have included a point-to-point response to your comments. We are also ready to further improve the manuscript if any extra comments are made. Detailed responses to reviewers are given below.
Thanks again for your patient help.
All the best,
Xiaoyan Cao
Reviewer 2: Comments and Suggestions for Authors:
The manuscript presented describes a novel member of the MYB transcription factors family from S. baicalensis. Authors characterized the gene ant its impact into two main flavonoids of S. baicalensis biosynthesis.
Point 1. Figure 1 presents levels of expression in different parts of the S. baicalensis plant. From the figure and correspondent description in the text it is not exactly clear xylem, phloem and periderm are the part of what roots or stem?
Response 1: I am very sorry for the inconvenience brought to you. In this article, xylem, phloem and periderm represent the part of roots, and the correspondent description has been corrected in the text in the revised manuscript (Line 121-122 in the revised manuscript).
Point 2. Also analyzing results presented in figure 2 authors concluded maximal expression levels of SbMYB12 in aboveground part of plant, especially leaves. But usually roots serves as a medicinal raw material for Scutellaria species. And what about concentration of flavonoids, especially baicalin and wogonoside, in native plants? What part of the plant accumulate maximal concentration of flavonoids?
Response 2: Thank you very much for your comment. The question you raised is very good. In native plants the concentration of flavanoids, especially baicalin and wogonoside are accumulated in a large amount in the root of Scutellaria baicalensis Georgi. Therefore, traditional Chinese medicine uses the dried root of S. baicalensis as medicine. At present, the contents of baicalin and wogonoside are used as the evaluation standard for the quality of S. baicalensis, but baicalin is the most important indicator.
Point 3. The data on expression profiles of genes supposed to be regulated with SbMYB12 overexpression are obtained on hairy roots model system. So, what is about a sustainability these expression pattern of baicalin synthesis in leaves, steam and roots of intact plant?
Response 3:Thank you very much for your comment. The problem you pointed out is very good, which is also the topic we want to further study. in S. baicalensis, only the hair root system can be used for transgenic operation at present, and the transgenic system for regenerating seedlings of complete plants has not been established, so we cannot know the physiological function of SbMYB12 in leaves and steam nowadays. Subsequently, if there are mature gene manipulation methods later, we can detect the regulation mode of SbMYB12 in stems, flowers and leaves.
Point 4. A scheme of flavonoid biosynthesis presented in figure 7 is based on authors own of literary data?
Response 4: Thank you very much for your comment. the scheme of flavonoid biosynthesis presented in figure 7 was based on the previous research achievements [Zhao, Q., et al. (2019). The Reference Genome Sequence of Scutellaria baicalensis Provides Insights into the Evolution of Wogonin Biosynthesis. Molecular plant, 12(7), 935–950. Hu, S. et al. (2022). Whole genome and transcriptome reveal flavone accumulation in Scutellaria baicalensis roots. Frontiers in plant science, 13, 1000469.]. And then we plotted the scheme of flavonoid biosynthesis and in this study.
Thanks again for your patient help.

Reviewer 3 Report
General comments:
This manuscript focuses on role of R2R3-MYB Transcription Factor SbMYB12 for baicalin biosynthesis in Scutellaria baicalensis Georgi. The SbMYB12 transcripts were expressed in all tested tissues (mainly in leaves) and responded to exogenous hormone methyl jasmonate (MeJA) treatment. The manuscript holds scientific potential and may be a novel approach to the pivotal role of SbMYB12 in baicalin biosynthesis. However, some major points and some typographical errors needs to be addressed before publication to overall enhance the quality of the manuscript. A moderate English Editing is must require to improve the quality of the manuscript.
Minor comments for authors:
Abstract: The authors should write about the findings of the present study rather than general statements.
L50-51: Rectify spacing error.
L58,87,94,99,105: The gene name should be in italics. Kindly correct it throughout the manuscript.
In Fig. 6 and 7, statistical analysis is not performed. Add statistical method.
Conclusion should be more precise. Highlight the major findings of the present study.
Some typographical errors are need to be addressed.

Author Response
Responses to Reviewer 3’s comments
Dear reviewers:
On behalf of all co-authors, I’d like to express our sincere gratitude to you for your time and helpful comments / suggestions on the earlier version of our manuscript (Manuscript ID: ijms-2002576) titled “A Novel R2R3-MYB Transcription Factor SbMYB12 Positively Regulates Baicalin Biosynthesis in Scutellaria baicalensis Georgi”. Your comments are very objective, which is very helpful for the improvement of our manuscript. Along with the revised manuscript with the “Track Changes”, we have included a point-to-point response to your comments. We are also ready to further improve the manuscript if any extra comments are made. Detailed responses to reviewers are given below.
Thanks again for your patient help.
All the best,
Xiaoyan Cao
Reviewer 3: Comments and Suggestions for Authors:
General comments:
This manuscript focuses on role of R2R3-MYB Transcription Factor SbMYB12 for baicalin biosynthesis in Scutellaria baicalensis Georgi. The SbMYB12 transcripts were expressed in all tested tissues (mainly in leaves) and responded to exogenous hormone methyl jasmonate (MeJA) treatment. The manuscript holds scientific potential and may be a novel approach to the pivotal role of SbMYB12 in baicalin biosynthesis. However, some major points and some typographical errors needs to be addressed before publication to overall enhance the quality of the manuscript. A moderate English Editing is must require to improve the quality of the manuscript.
Minor comments for authors:
Point 1. Abstract: The authors should write about the findings of the present study rather than general statements.
Response 1: Thank you very much for your comment. According to your suggestion, we have rewritten the Abstract part in the revised version.
Point 2. L50-51: Rectify spacing error.
Response 2: I am really sorry for the inconvenience and thank you for pointing out the problem. The correct information has been revised in the revised manuscript (Line 53-54 in the revised manuscript).
Point 3. L58,87,94,99,105: The gene name should be in italics. Kindly correct it throughout the manuscript.
Response 3: I am really sorry for the inconvenience and thank you for pointing out the problem. We have checked the full text and the correct information has been revised in the revised manuscript.
Point 4. In Fig. 6 and 7, statistical analysis is not performed. Add statistical method.
Response 4: Thank you very much for your comment. We have added statistical method in Fig. 6 and 7 in the revised manuscript (Line 216-218, Line 235-236 in the revised manuscript).
Point 5. Conclusion should be more precise. Highlight the major findings of the present study.
Response 5: Thank you very much for your comment. Your suggestions are of great help to improve the level of our article. We have highlighted the major findings of the present study in the Conclusion of the revised manuscript.
Point 6. Some typographical errors are need to be addressed.
Response 6: Thank you very much for your comment. We have carefully reviewed the full text and addressed the typographical errors.
Thanks again for your patient help.

Reviewer 4 Report
This study tried to demonstrate that SbMYB12 involved in flavonoid biosynthesis. However, there are many weaknesses in this manuscript.
1. Full-name of MYB should be shown for the first time in the manuscript.
2. Protein expression levels of SbMYB12 in the different tissues are much important than their mRNA levels, it is better to show the protein levels in these tissues. It is the most important information to support the conclusion that the transcription factor really promotes the biosynthesis pathway in the leaf tissue. 3. For the subcellular localization experiment, the authors used “Arabidopsis” mesophyll protoplasts to demonstrate the nuclear localization. Why? Do any references can support that is convincing for the replacement. 4. SbMYB12 is mainly expressed in the leaf not in the root. Why the authors used hairy roots to demonstrate the biosynthesis of flavonoids? It may be totally different situations in the two tissues. 5. As shown in Figure 7, there are more than three genes unregulated, but only the promoters of three genes were examined. Why? 6. In Figure 8A, the third gene is SbF6H-1, however, the third gene is SbF6H-2 in the context. Please check it out. 7. In the Line 300, it is better to add “some of” before S. Baicalensis-specific flavonoid.Author Response
Responses to Reviewer 4’s comments
Dear reviewers:
On behalf of all co-authors, I’d like to express our sincere gratitude to you for your time and helpful comments / suggestions on the earlier version of our manuscript (Manuscript ID: ijms-2002576) titled “A Novel R2R3-MYB Transcription Factor SbMYB12 Positively Regulates Baicalin Biosynthesis in Scutellaria baicalensis Georgi”. Your comments are very objective, which is very helpful for the improvement of our manuscript. Along with the revised manuscript with the “Track Changes”, we have included a point-to-point response to your comments. We are also ready to further improve the manuscript if any extra comments are made. Detailed responses to reviewers are given below.
Thanks again for your patient help.
All the best,
Xiaoyan Cao
Reviewer 4: Comments and Suggestions for Authors:
This study tried to demonstrate that SbMYB12 involved in flavonoid biosynthesis. However, there are many weaknesses in this manuscript.
Point 1. Full-name of MYB should be shown for the first time in the manuscript.
Response 1: Thank you very much for your comment. We have indicated in the revised version the full biological name "MYB" at the beginning of the sentence in the manuscript (Line 47 in the revised manuscript).
Point 2. Protein expression levels of SbMYB12 in the different tissues are much important than their mRNA levels, it is better to show the protein levels in these tissues. It is the most important information to support the conclusion that the transcription factor really promotes the biosynthesis pathway in the leaf tissue.
Response 2: Thank you very much for your comment. We think that the question you raised is very scientific and meaningful, but so far, there is no commercial antibody for SbMYB12, leading to no mature system for protein level detection. If possible, we will prepare S. baicalensis related protein antibody for protein level detection.
Point 3. For the subcellular localization experiment, the authors used “Arabidopsis” mesophyll protoplasts to demonstrate the nuclear localization. Why? Do any references can support that is convincing for the replacement.
Response 3: Thank you very much for your comment. As a model plant, Arabidopsis thaliana plays an important role and has great advantages in biological research. There have been many reports shown that the subcellular localization experiment of heterologous genes has been successfully verified by transient expression of Arabidopsis protoplasts [Yoo, S. D., et al. (2007). Arabidopsis mesophyll protoplasts: a versatile cell system for transient gene expression analysis. Nature protocols, 2(7), 1565–1572.]. Therefore, we selected mature Arabidopsis protoplast system to verify the subcellular localization of SbMYB12.
Point 4. SbMYB12 is mainly expressed in the leaf not in the root. Why the authors used hairy roots to demonstrate the biosynthesis of flavonoids? It may be totally different situations in the two tissues.
Response 4: Thank you very much for your comment. Our study found that SbMYB12 is expressed in all tissues, and it was found in our previous study that it can actively respond to exogenous MeJA treatment. Moreover, according to the results of MYB phylogenetic analysis, we speculated that it is involved in the synthesis and regulation of flavonoids, so we took it as our key gene to study. However, the genetic transformation system of regenerated Scutellaria baicalensis seedlings has not been established yet, so we had to use the mature hair-root genetic transformation system to study the function of this gene, and found that it can promote the accumulation of flavonoids in the transgene hairy root of S. baicalensis. Interestingly, the expression level of SmMYB97 in roots was significantly lower than that in stems, leaves and flowers, but it could positively regulate the expression of salvianolic acid and other flavonoids in the roots of Salvia miltiorrhiza [Li, L., et al. (2020). JA-Responsive Transcription Factor SmMYB97 Promotes Phenolic Acid and Tanshinone Accumulation in Salvia miltiorrhiza. Journal of agricultural and food chemistry, 68(50), 14850–14862.]. The expression level of SmSPL7 in leaves was significantly higher than that in roots, and it could positively regulate the synthesis of anthocyanin flavonoids [Chen, R., et al. (2021). Transcription factor SmSPL7 promotes anthocyanin accumulation and negatively regulates phenolic acid biosynthesis in Salvia miltiorrhiza. Plant science, 310, 110993.]. These studies in Salvia miltiorrhiza have the same significance as our findings, suggested that these genes have larger functions to be studied in the future.
Point 5. As shown in Figure 7, there are more than three genes unregulated, but only the promoters of three genes were examined. Why?
Response 5: Thank you very much for your comment. At that time, we cloned the promoter sequence of the enzyme genes of baicalin synthesis pathway, including these three interacting genes(SbCLL7-4, SbCHI-2, and SbF6H-1), and found that only these three genes(SbCLL7-4, SbCHI-2, and SbF6H-1) can interact with SbMYB12 through the screening of Yeast one-hybrid. So, we only included three interacting genes in this paper
Point 6. In Figure 8A, the third gene is SbF6H-1, however, the third gene is SbF6H-2 in the context. Please check it out.
Response 6: I am really sorry for the inconvenience and thank you for pointing out the problem. The correct information has been revised in the revised manuscript. The correct genes to be cloned is SbF6H-1(Line 242-243 in the revised manuscript).
Point 7. In the Line 300, it is better to add “some of” before S. Baicalensis-
Response 7: Thank you very much for your comment. We have added “some of” before S. Baicalensis- in the revised version (Line 305 in the revised manuscript).
Thanks again for your patient help.

Round 2
Reviewer 2 Report
Dear authors!
Thank for the detailed responce. Wish you all the best in future investigations and publishing the manuscript as soon as it possible!
Reviewer 4 Report
Accept in present form.